# Manuka Honey Reduces NETosis on an Electrospun Template Within a Therapeutic Window

**DOI:** 10.3390/polym12061430

**Published:** 2020-06-26

**Authors:** Benjamin A. Minden-Birkenmaier, Richard A. Smith, Marko Z. Radic, Marie van der Merwe, Gary L. Bowlin

**Affiliations:** 1Department of Biomedical Engineering, University of Memphis, 330 Engineering Technology Building, Memphis, TN 38152, USA; bmndnbrk@memphis.edu; 2Department of Orthopaedic Surgery & Biomedical Engineering, University of Tennessee Health Science Center, E228A Coleman Building, 956 Court Avenue, Memphi, TN 38163, USA; rsmith40@uthsc.edu; 3Department of Microbiology, Immunology and Biochemistry, University of Tennessee Health Science Center, 201 Molecular Science Building, 858 Madison Ave., Memphis, TN 38152, USA; mradic@uthsc.edu; 4School of Health Studies, University of Memphis, Fieldhouse 310, Memphis, TN 38152, USA; mvndrmrw@memphis.edu

**Keywords:** Manuka honey, NETosis, inflammation, neutrophil, electrospinning, tissue engineering, polydioxanone

## Abstract

Manuka honey, a topical wound treatment used to eradicate bacteria, resolve inflammation, and promote wound healing, is a focus in the tissue engineering community as a tissue template additive. However, its effect on neutrophil extracellular trap formation (NETosis) on a tissue engineering template has yet to be examined. As NETosis has been implicated in chronic inflammation and fibrosis, the reduction in this response within the wound environment is of interest. In this study, Manuka honey was incorporated into electrospun templates with large (1.7–2.2 µm) and small (0.25–0.5 µm) diameter fibers at concentrations of 0.1%, 1%, and 10%. Template pore sizes and honey release profiles were quantified, and the effect on the NETosis response of seeded human neutrophils was examined through fluorescence imaging and myeloperoxidase (MPO) analysis. The incorporation of 0.1% and 1% Manuka honey decreased NETosis on the template surface at both 3 and 6 h, while 10% honey exacerbated the NETosis response. Additionally, 0.1% and 1% Manuka honey reduced the MMP-9 release of the neutrophils at both timepoints. These data indicate a therapeutic window for Manuka honey incorporation into tissue engineering templates for the reduction in NETosis. Future in vivo experimentation should be conducted to translate these results to a physiological wound environment.

## 1. Introduction

The implantation of a biomaterial tissue engineering template into the body initiates an orchestrated series of events which have important ramifications on the ultimate success or failure of that template. Immediately, proteins in the pooled blood around the template begin to adsorb to its surface, altering its properties [1]. Inflammatory signals released in response to the implantation injury recruit immune cells which interact with the template and release a host of cytokines and chemokines, recruiting more immune cells and setting the area down a path towards either desirable tissue integration/regeneration or undesirable fibrosis [2]. The first wave of these immune cells consists of neutrophils, fast-arriving phagocytic cells which exert control over the inflammatory state of the wound through the release of cyto/chemokines [3]. As the first-responders, these cells play a key role in the preconditioning of the template surface and surrounding microenvironment that influences the ultimate fate of the tissue engineering template.

In addition to their roles as a phagocyte and a producer of cyto/chemokines, neutrophils also have the ability to eject a mixture of DNA and bactericidal granular components to create neutrophil extracellular traps (NETs) during a cell death process known as NETosis [4]. This process is triggered by bacterial signals such lipopolysaccharide (LPS) and formylated peptides such as formyl-met-leu-phe (fMLP), as well as inflammatory stimuli such as interleukin 8. The intracellular responses to these signals increase the production of intracellular reactive oxygen species, which then activate an enzyme known as PAD4 to citrullinate histone H3 [5]. Citrullination of these histones causes the chromatin to decondense, which then leads to the collapse of the nuclear membrane, adsorption of granular components by the chromatin, rupture of the plasma membrane, and finally NET release. This response is a key weapon against pathogens, trapping them to reduce the spread of infection and killing them with bactericidal granule components such as neutrophil elastase (NE) [6]. It has also been demonstrated that neutrophils undergo NETosis in response to large foreign particles that they cannot phagocytose, such as urate crystals, polystyrene beads, and cholesterol crystals [7,8,9]. Accordingly, the effect of nanofibrous template architecture, specifically fiber diameter and corresponding pore size, on NETosis has been recently examined [10]. It was demonstrated that templates with small polydioxanone (PDO) fiber diameters (0.3 ± 0.1 µM) (SD) and pore sizes (1.0 ± 0.3 µM) trigger a significant increase in NETosis relative to templates with large fiber diameters (1.9 ± 1.0 µM) (LD) and pore sizes (8.1 ± 3.6 µM). Furthermore, the amount of NETosis on these templates was correlated with fibrous capsule formation around the template in vivo. As such, it is of interest to develop methods of regulating NETosis on the surface of these tissue engineering templates to improve tissue-template integration and subsequent tissue regeneration.

One possible method of regulating NETosis is through the incorporation of Manuka honey into electrospun nanofibrous templates. Manuka honey, a variety of honey produced in New Zealand from the nectar of the leptospermum scoparium shrub, is a wound treatment that has recently become the subject of investigation as a tissue engineering template additive [11,12,13]. The honey’s methylglyoxal content gives it potent antibacterial activity against a wide range of bacteria, including antibiotic-resistant bacteria [14,15,16,17]. Additionally, the honey contains flavonoids which scavenge tissue-damaging free oxygen radicals, and particular phenolic components have been demonstrated to activate intracellular antioxidant response pathways [18,19]. When incorporated into a nanofibrous template, Manuka honey increases the proliferation and infiltration of fibroblasts [11]. Recently, we also examined the effect of Manuka honey on a dHL-60 neutrophil model, demonstrating that Manuka honey reduces chemotaxis, reduces superoxide production and the activation of the pro-inflammatory nuclear factor κB (NF-κB) intracellular signaling pathway, and reduces matrix metalloproteinase production in response to pro-inflammatory signals [20,21,22]. Given its use in tissue engineering templates and its promising ability to regulate other neutrophil inflammatory behaviors, the logical next step is to evaluate the ability of Manuka honey-laden tissue templates to modulate neutrophil NETosis.

In the current study, Manuka honey was incorporated into electrospun PDO tissue engineering templates with small (SD) (0.25–0.50 µM) and large (LD) (1.7–2.1 µm) fiber diameters. These templates were characterized with regard to morphology and honey release. Primary human neutrophils were cultured on the templates for 3 and 6 h, and their degree of NETosis on the surface of these templates was characterized using fluorescence imaging and a myeloperoxidase (MPO)-based MAGPIX procedure. The release of interleukin 8 (IL-8/CXCL8), interleukin 1-β (IL-1β), interleukin 10 (IL-10), interleukin 1 receptor antagonist (IL-1ra), tumor necrosis factor α (TNF-α), and matrix metalloprotinease 9 (MMP-9) from these neutrophils was also quantified using a multiplexed immunoassay. We hypothesized that Manuka honey incorporation would reduce the degree of NETosis, especially on the NETosis-provoking SD templates, and that Manuka honey would reduce IL-8/CXCL8, IL-1β, MMP-9, and TNF-α release while increasing IL-10 and IL-1ra release from the neutrophils.

## 2. Materials and Methods 

### 2.1. Electrospinning

Manuka honey (UMF12+, Manuka Guard) was added to 1,1,1,3,3,3-hexafluoro-2-propanol (HFP) (Oakwood Products, Estill, SC, USA) solutions at concentrations of 0.1, 1, or 10% *v/v*, allowed to dissolve at 37 °C for 5 min, then vortexed to disperse. PDO (Bezwada Biomedical, Hillsborough, NJ, USA) was then added to the honey/HFP solutions and allowed to dissolve overnight (samples were also formulated from pure HFP solutions as non-honey controls). PDO concentrations were 67 mg/mL PDO for SD templates or 135–140 mg/mL PDO for LD templates. Solutions were electrospun from 3 mL syringes (Becton, Dickinson and Company, Franklin Lakes, NJ, USA) through either an 18-gauge blunt needle (LD templates) or a 26 gauge needle (SD templates) attached to a positive power source (Spellman CXE1000R, Spellman High Voltage Electronics, Hauppauge, NY, USA) using a syringe pump (Model 78-01001, Fisher Scientific, Waltham, MA, USA). Specific polymer concentrations, flow rates, air gap distance, and applied voltage parameters for low humidity conditions are displayed in Table 1 and for high humidity conditions in Table 2. Parameters for certain templates were varied due to atmospheric humidity fluctuations to maintain a consistent fiber diameter for each template type. Electrospun fibers were collected onto a stainless steel grounded rectangular mandrel (20 × 75 × 5 mm) horizontally translating 6.5 cm/s over a range of 13 cm and vertically rotating at 1250 rpm. Three milliliters of the solution was spun for the SD templates, and 2 mL of the solution was spun for the LD templates to equalize template thickness. The templates, which ranged in thickness from 0.05 to 0.15 mm, were stored at −20 °C prior to use.

### 2.2. Template Characterization

To characterize fiber diameters and pore sizes, samples of each template were sputter coated with 5 nm gold-palladium under argon and imaged using SEM imaging. Images were taken via a FEI Nova NanoSEM (FEI, Hillsboro, OR, USA) 650 at +20 kV with a spot size of 3 and a 5 mm working distance. Fiber diameters were measured via FibraQuant 1.3 software (nanoTemplate Technologies, LLC, Chapel Hill, NC, USA), and mean fiber diameters and standard deviations were calculated from a minimum of 200 semi-automated random measurements per image. Image J software (NIH, Bethesda, MD, USA) was used to measure pore diameters from a minimum of 3 SEMs per template type, with a minimum of 60 measurements per template type.

### 2.3. Honey Release

To characterize the honey release profile of each template type, 8 mm diameter circular punches were placed in a 96-well plate with 150 µL of Hank’s Balanced Salt Solution (HBSS) (Gibco) and incubated at 37 °C. At timepoints of 15, 30, 45 min, 1, 2, 3, 6, 12 h, and 1, 2, 3, 7, 14, and 21 days, HBSS was removed from each well and saved for analysis, and 150 µL of new HBSS was added. Collected releasate was stored at −20 °C until analysis. A spectral scan of Manuka honey diluted in HBSS was performed via a SpectraMax i3x Multi-Mode Microplate Reader, and it was determined 340 nm was the optimum wavelength for honey quantification. A serial dilution of Manuka honey was then created and its absorbance was read at 340 nm via a SpectraMax i3x Multi-Mode Microplate Reader (Molecular Devices, San Jose, CA, USA) to establish a standard curve. The collected releasate from the experimental samples was then read for absorbance at 340 nm, and the standard curve was used to calculate the volumetric percentage of Manuka honey in each sample. Four template punches of each template type were used.

### 2.4. Response of Primary Human Peripheral Blood Neutrophils

Eight-millimeter diameter circular punches of each template type were disinfected by irradiating with UV light using a Spectroline lamp (8 watts, Part No. EN280L, Spectroline, Westbury, NY, USA) at a working distance of 9.5 cm between the lamp and the punches. UV light with a 365 nm wavelength was applied for 10 min on each side in a sterile laminar flow hood. Healthy donor blood was obtained from Tennessee Blood Services and de-identified prior to use as specified in approved IRB #4234 (exempt) by the University of Memphis Institutional Review Board, and neutrophils were isolated as described in a previous publication [5]. Once isolated, neutrophils were resuspended in HBSS without calcium and magnesium but with sodium bicarbonate, 10 mM HEPES (Gibco) and 0.2% autologous serum (hereafter referred to as HBSS+). The disinfected template punches were placed in a 96-well plate with the non-mandrel-contacting surface facing up, and 40 µL HBSS+ was added to each well to hydrate the templates. Then 100,000 neutrophils were added in 100 µL HBSS+ to each well, and 10 µL of 150 U/mL heparin (Sigma Aldrich, St. Louis, MO, USA) was added to each well (final concentration of 10 U/mL) to dissociate NET-associated MPO for the quantification of NETs as indicated in the literature [23]. Replicates of blanks and 10% honey samples were run without the heparin added to check that non-NET-associated MPO was negligible. Templates were cultured at 37 °C at 5% CO_2_ for 3 or 6 h. Tissue culture plastic (TCP) positive and negative controls with 100 nM phorbol 12-myristate 13-acetate (PMA) (Sigma Aldrich, St. Louis, MO, USA) or vehicle (0.15% dimethylsulfoxide, Fisher Scientific, Waltham, MA, USA) were run in parallel with the same amount of cells. Following the incubation period, samples were placed on ice for 10 minutes to inhibit neutrophil stimulation. Supernatants were saved and frozen for later analysis, and templates were fixed with 10% buffered formalin (Fisher Scientific, Waltham, MA, USA) for one hour at room temperature. After a 5-min wash with phosphate buffered saline (PBS) (Hyclone, GE Healthcare Life Sciences, Marlborough, Massachusetts, USA), samples were stored in PBS at 4 °C until further use. Four template punches of each template type were used per timepoint. 

### 2.5. Quantifying NETosis Response via Fluorescent Imaging

Templates were immunostained for NE. First, free aldehyde groups were quenched with 3 washes of 25 mM glycine (Fisher Scientific) in PBS for 5 min each. Then templates were blocked with 5% milk in PBS for 1 h. Then, rabbit anti-human NE antibody (Abcam, ab21595) was reconstituted at a 1:100 dilution and incubated on the templates for an hour. After 3 washes with 0.1% tween in PBS for 5 min each, goat anti-rabbit IgG H&L AlexaFluor 488 (Abcam, ab15007) was reconstituted at a 1:200 dilution and incubated on the templates for an hour. This was followed by two washes with 0.1% tween (Fisher Scientific, Waltham, MA, USA) and one wash with PBS, all for five minutes. All of these steps were completed at room temperature with gentle agitation. Then, extracellular DNA was stained on each template with 5 µM sytox orange (SO) (Invitrogen, cat. no. S34861, Carlsbad, CA, USA) in deionized water for 5 min. Intact nuclei were then stained with 4’6-diamidino-2-phenylindole (DAPI) (Life Technologies, cat. no. R37606, Carlsbad, California, USA) for 5 min at stock concentration. After one more 5-minute PBS wash, imaging was undertaken with an Olympus BX43F microscope with an Olympus DP73 high-performance digital color camera and an Olympus U-HGLGPS fluorescent light source (Olympus, Shinjuku City, Tokyo, Japan). Three representative images at 20× magnification were taken of each template punch, and 4 template punches of each type were imaged per timepoint during each experiment for three experiments with three unique donors for a total of 36 images per template type per timepoint. The background was subtracted by defining a region of interest containing no NETs. CellSens^®^ Standard 1.9 Digital Imaging software (Olympus, Shinjuku City, Tokyo, Japan) was used to record images, and the exposure time and fluorescence settings for SO, NE, and DAPI were held constant for all images. Using a custom MATLAB^®^ program (MATLAB R2019b, Mathworks, Natlick, MA, USA), the percent area covered by red pixels but not blue was calculated (percent area NETs). The number of viable cells (blue/red ratio of 1:3 or greater), number of NETosing cells (colocalization of blue, red, and green pixels), and the number of necrotic cells (blue/red ratio of 1:3 or less) were calculated using size thresholding and circularity to identify discrete cells. Three experiments using neutrophils from three age-matched (age range 22–28 years old) and gender-matched (male) donors were performed. 

### 2.6. MAGPIX Assay

Supernatants from the neutrophil experiments were assayed using a multiplexed magnetic bead immunoassay (R&D Systems, Minneapolis, MN, USA) on a MAGPIX^®^ reader (Luminex Corp., Austin, TX, USA) for MPO, TNF-α, MMP-9, IL-8, IL-10, IL-1β, and IL-1ra. Samples were run using a dilution factor of 50 in assay diluent.

### 2.7. Statistical Methods

Replicate numbers for each experiment are indicated in the figure captions throughout the text. A Shapiro–Wilk test was used to determine the normality of each data set. Where the data were found to follow a normal distribution for experiments with six replicates, a one-way or two-way analysis of variance (ANOVA) was used with a Holm–Sidak post hoc test to determine significance (α = 0.05). Where the data were found to be non-normal, or the experiment had fewer than six replicates, a Kruskal–Wallis ANOVA on ranks was used with a Mann–Whitney post hoc test to determine significance (α = 0.05). Analysis was performed in GraphPad Prism 6 (GraphPad Software Inc., San Diego, California, USA).

## 3. Results

Figure 1 displays SEM images of each template type, and Table 3 contains the FibraQuant measurements of fiber diameter and ImageJ measurements of pore diameter. As can be seen in these images, the solutions of low and high polymer concentration created templates of small and large fiber diameters, respectively. The honey did not adversely affect fiber morphology, and fiber diameters were comparable between all SD and LD templates regardless of honey concentration. However, increasing honey concentration did decrease pore diameter significantly in the SD templates as compared to the non-honey blank (non-significant trend in the LD templates), suggesting that the honey caused tighter fiber packing and/or fiber melding. All SD templates had significantly smaller pore diameters than the corresponding LD template. Figure 2 contains histograms of the pore distributions in each template type. As displayed in the figure, in the absence of honey, the LD templates have about 60% of their pores with diameters greater than 10 µm, allowing for infiltration of neutrophils (8.5 µm in diameter) [24]. In contrast, the SD templates without honey have only about 20% of their pores with diameters greater than 10 µm. Although honey shifts the pore distribution towards smaller pores, all LD templates with honey have at least 25% of their pores above 10 µm in diameter, while the SD templates with honey have a negligible number of pores with diameters above 10 µm, indicating that neutrophil infiltration would likely be slowed or blocked in these templates. Figure 3 contains the honey release results for each template type over a 21-day release period. As shown in the graphs, most of the honey is released as a burst within the first several hours of hydration, but there is a low-level consistent release of honey from the templates throughout the entire 21-day period. The 10% honey SD template released significantly more honey than the 10% honey LD template at each timepoint. This can be explained by the greater surface-area-to-volume ratio of the smaller fibers, providing more surface area for honey release than the LD fibers. The 0.1% honey SD template also had greater honey release than the 0.1% honey LD template, although this difference was not statistically significant. In general, the released amount of honey was proportional to the amount of honey electrospun into the template with each template type. 

Representative fluorescence images of NETosis on SD and LD templates at 3 and 6 h are displayed in Figure 4 (all images taken from NETosis experiment 3). Red (SO) marks the extracellular DNA indicative of NETosis, while blue (DAPI) indicates the intact nuclei of non-NETosing neutrophils and green indicates NE (colocalized with red-stained DNA in NETs, appearing as yellow as the NET is released). These images demonstrate the higher degree of NETosis found on the SD blank templates relative to the corresponding LD blank templates at both 3 and 6 h, as expected. These images also show the decrease in NETosis on the SD 0.1% honey templates relative to the SD blank templates at 3 and 6 h, and the increase in NETosis on the 10% honey templates in both the LD and SD samples. Figure 5 contains the quantified percent area NETs for each template type at 3 and 6 h compiled from three experiments with neutrophils from three unique donors. As expected, the SD blank samples had significantly more percent area NETs than the LD blank templates at both 3 and 6 h. The SD 0.1% honey templates also had significantly less percent area NETs than the SD blank templates at both timepoints. Meanwhile, 10% honey significantly increased percent area NETs in both the LD and SD templates at 6 h, and had a non-significant increasing trend in the LD templates at 3 h. Together, these results demonstrate that 0.1% honey incorporation is effective at reducing neutrophil NETosis on the surface of SD templates, but that the incorporation of 10% honey induces significant NETosis on the surface of both template types. Figure 6 contains the percent NETosing and necrosing nuclei for each template type at each timepoint. As demonstrated by the data, there was no significant trend in the percentage of necrotic cells for either of the template types at either timepoint, indicating that the honey is not causing necrosis. There is a significant increase in NETosing cells on the LD 10% honey samples at both timepoints, correlating with the increase in percent area NETs observed in Figure 5. There are no significant differences in NETosing cells in the SD samples at 3 h, but at 6 h, there is a significant decrease in NETosing cells in the SD 0.1% honey samples and SD 10% honey samples. The decrease at 0.1% honey correlates with a decrease in percent area NETs in these samples, while the decrease at 10% honey may be due to the fact that these cells have already progressed in their NETosis response to the extent that their nuclei are no longer identifiable. Figure 7 contains the MPO concentrations in the supernatants taken from LD and SD templates with and without 10% honey cultured in the presence or absence of heparin. This experiment was performed to ensure that the amount of non-NET-associated MPO was low enough that it would not confound the ability of the MPO assay to accurately quantify NETosis. As the amount of MPO measured in the supernatants of the non-heparin-treated samples was negligible relative to the heparin-treated samples, this experiment determined that the vast majority of MPO available was NET-bound heparin. Thus, the ability of the MPO assay to serve as an accurate measurement of the NETosis response was validated.

The MPO quantification of each sample type at 3 and 6 h are displayed in Figure 8, normalized to each experiment’s 6-h TCP 100 nM PMA positive control and expressed as the percent of the NETosis response. The 1% honey LD template punches had a significant decrease in the percent NETosis response at 3 h relative to the LD blank template samples, and both the 0.1% and 1% honey samples had significantly less percent NETosis than the LD 10% samples at this timepoint. Similarly, the 0.1% and 1% honey LD samples had significantly less percent NETosis than the LD 10% honey samples at 6 h. The 0.1% honey samples also had significantly less NETosis than the 10% honey samples in the SD templates at 3 h, although this difference was not significant at 6 h. Together, these results confirm the data presented in Figure 5 which indicated that 0.1% and 1% honey suppressed NETosis while 10% honey exacerbated NETosis. It should be noted that minor differences in which groups are statistically significant from each other between Figure 5 and Figure 8 can be explained by the different components being measured (i.e., DNA and MPO) and the different measurement mechanisms (image analysis and immunoassay). The imaging method has additional variability stemming from the unequal distribution of NETs across the template surface, as well as the use of a 2-D image system to measure 3-D NET structures.

Although the sample supernatant was assayed for TNF-α, MMP-9, IL-8, IL-10, IL-1β, and IL-1ra, only MMP-9 was found in non-negligible amounts. The MMP-9 levels, displayed in Figure 9, loosely track the percent area NETs and percent NETosis response results. The SD and LD blanks had similar levels of MMP-9 release. The 1% honey LD samples had significantly less MMP-9 release than the LD blank samples at 3 h, and at 6 h the 0.1% honey LD samples had significantly less MMP-9 than the blank samples, while both 0.1% and 1% honey had significantly less MMP-9 than the LD 10% honey samples. Similarly, at 3 h, the 0.1% honey SD samples had less MMP-9 than the SD 1% and SD 10% honey samples, and the 0.1% honey samples had significantly less MMP-9 release than the blanks. At 6 h, SD 0.1% honey had significantly less MMP-9 than the 1% and 10% honey samples. Together, these results demonstrate that the incorporation of 0.1% and 1% honey reduces MMP-9 production by neutrophils in contact with LD templates, while only 0.1% honey had a similar effect on the SD templates. Note that that unlike the MPO assay displayed in Figure 8, the MMP-9 results in Figure 9 do not measure a NET component, but rather a soluble cytokine released by neutrophils into the cell culture media. As such, these data are not an indicator of NETosis, but rather a concomitant neutrophil inflammation response. The lack of TNF-α, IL-8, IL-1β, IL-10, and IL-1ra release by neutrophils seeded on these templates indicate that they are not activated into pro-inflammatory or anti-inflammatory behavior. 

## 4. Discussion

Because of its potent properties as a wound-healing agent, Manuka honey has become a common additive for tissue engineering templates, including electrospun templates, cryogels, and hydrogels [11,12,25]. Despite this predominance in the field, no one has yet examined the effect of Manuka honey incorporation on neutrophil NETosis at the surface of the tissue template. As neutrophils are one of the first inflammatory cell types to interact with an implanted material, and the NETosis response is an important predictor of the degree of fibrosis surrounding this material, assessing this response is crucial to predicting the success or failure of honey-laden tissue engineering templates. When incorporated into electrospun templates, it has previously been reported that the incorporation of Manuka honey reduces pore size, and the results shown in Figure 1 and Figure 2 confirm this finding [11]. It has been theorized that restrictive porosities in electrospun templates increase the degree of neutrophil NETosis on the surface by preventing neutrophils from infiltrating into the template interior, causing them to perceive the material as an impenetrable surface rather than an extracellular matrix analogue [10]. This theory has been suggested as an explanation for why SD templates exhibit more NETosis than their LD counterparts. Similarly, the decrease in pore size with increasing honey content could contribute to the corresponding increase in NETosis on the 10% honey templates. However, the 10% honey LD templates have a greater proportion of pores above 10 µm in diameter than the SD blank templates but exhibit similar levels of NETosis. Furthermore, the 0.1% SD honey templates have a lower proportion of pores above 10 µm in diameter than the SD blank templates but have significantly lower percent area NETs. As such, a decrease in porosity cannot explain the entirety of honey’s effect on neutrophil NETosis. 

An alternative explanation for the increase in NETosis in the 10% honey samples is that the glucose content of the honey triggers NETosis when honey is present at a high enough concentration. A Menegazzo et al. 2014 paper demonstrated that glucose could trigger NETosis on its own and exacerbate NETosis when triggered by another agonist [26]. As glucose is a major component of honey, it is possible that neutrophils contacting the surface of 10% honey-laden fibers encounter concentrations of glucose high enough to trigger their NETosis response [27]. At lower concentrations of honey, such as those encountered at the surface of a 0.1% or 1% honey template, this glucose level may not be high enough to trigger the NETosis response. As Manuka honey also contains antioxidants such as pinobanksin which have been theorized to cross the cell membrane and scavenge ROS, it is possible that these antioxidants disrupt the ROS-dependent mechanisms which trigger NETosis [28]. The interplay of the glucose and antioxidant stimulatory and inhibitory mechanisms could create a therapeutic concentration window in which NETosis is inhibited at the surface of a tissue engineering template, similar to the phenomena observed in Figure 5 and Figure 8. However, more research will have to be conducted to determine whether these proposed mechanisms are the cause of honey’s ability to reduce NETosis on a tissue engineering template when incorporated at concentrations of 0.1% and 1% but stimulate NETosis at a concentration of 10%. It has been previously reported that Manuka honey decreases the activation of the pro-inflammatory NF-κB pathway, but the specific components of the honey which cause this change have yet to be isolated [20,29]. Future work should also investigate whether Manuka honey affects intracellular signaling events which lead to NETosis, such as PAD4 activation or histone H3 citrullination. The ability of Manuka honey to modulate soluble MMP-9 release in a similar manner to its effect on NETosis provides an intriguing tool for the treatment of chronic wounds. It should be noted that the MMP-9 results and NETosis results measure two different phenomena (the release of soluble MMP-9 and the extrusion of NETs, respectively), and there are differences between the MMP-9 results and the NETosis results. For instance, MMP-9 release from neutrophils on LD templates was decreased by the incorporation of 0.1%–1% honey to a much greater degree than the decrease in NETosis on these templates. While MMP-9 is crucial to clearing debris and opening channels for vascular endothelial cells to create new blood vessels during the healing process of normal wounds, elevated levels of MMPs in chronically-inflamed wounds have been implicated in a constant cycle of tissue damage that precludes healing [30]. Accordingly, the ability of 0.1% and 1% honey-containing LD templates and 0.1% honey SD templates to reduce MMP-9 release from neutrophils could reduce this tissue damage when placed on a chronic wound, thus aiding in the resolution of inflammation and the resumption of healing activity. 

## 5. Conclusions

The degree of neutrophil NETosis on a tissue engineering template is predictive of the degree of fibrosis it will encounter in vivo, and therefore its success or failure to integrate with the surrounding tissue and induce regeneration. This study indicates that the incorporation of Manuka honey in the range of 0.1%–1% in a tissue engineering template decreases the NETosis response of neutrophils encountering that template, with a corresponding decrease in the MMP-9 release of these neutrophils. The data presented in this study demonstrate the importance of testing honey-laden biomaterials for their impact on the NETosis response. Given the widely divergent impact on NETosis that different honey levels have, it is vital that the surface properties and honey levels of such biomaterials be tailored to achieve the desired NETosis response in order to have the appropriate effect in a wound site or other applications. Future work will include correlating these findings with in vivo fibrosis studies and attempting to delineate the effect of Manuka honey itself on neutrophils from the effect of the restrictive porosities that it creates when incorporated into electrospun templates.

## Figures and Tables

**Figure 1 polymers-12-01430-f001:**
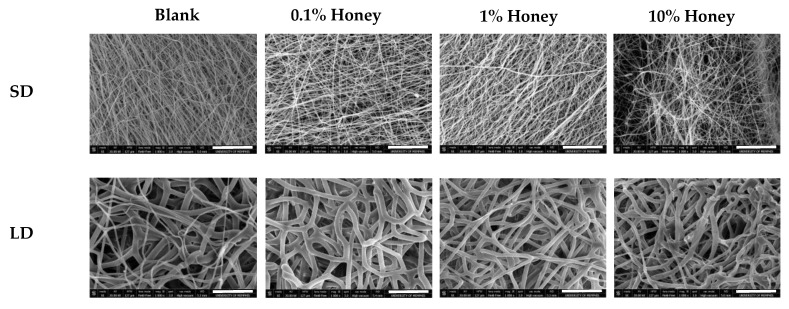
Low and high polymer concentrations create small diameter (SD) and large diameter (LD) fibers. SEM images of each template type. Scale bars = 30 µm.

**Figure 2 polymers-12-01430-f002:**
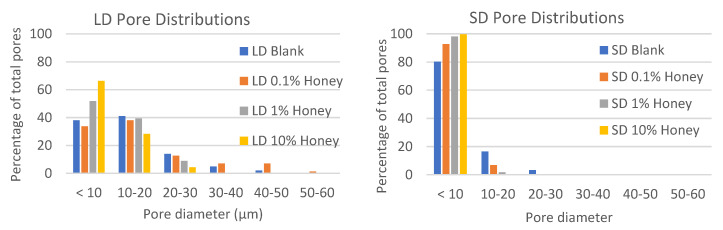
Honey reduces pore size. Histograms of pore distributions in LD and SD templates with 0%, 0.1%, 1%, and 10% honey.

**Figure 3 polymers-12-01430-f003:**
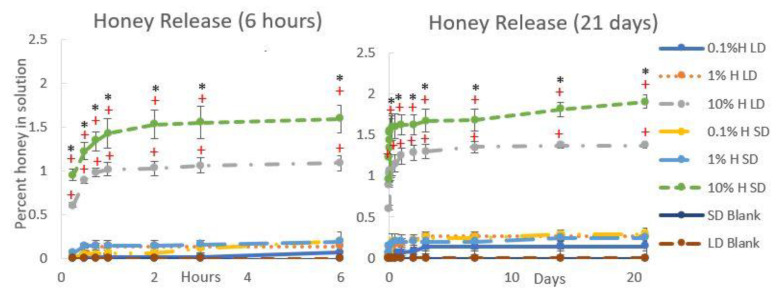
Honey release is higher for SD templates than LD templates. Percent honey *v/v* in PBS incubated with each template type for 21 days. * indicates a significant difference between the LD and SD templates of the corresponding honey concentration, and + indicates a significant difference from all other honey concentrations and non-honey blank of the corresponding fiber diameter (SD or LD). Note: the 1% honey LD samples are significantly different from the LD blank from day 3 onward, the 1% honey SD samples are significantly different from SD blank for days 14 and 21, and the 0.1% honey SD samples are significantly different from SD blank from day 3 onward. Due to clustering of these lines on the graphs, these differences were unable to be notated. Sample size = 4. Raw data is available in Appendix A.

**Figure 4 polymers-12-01430-f004:**
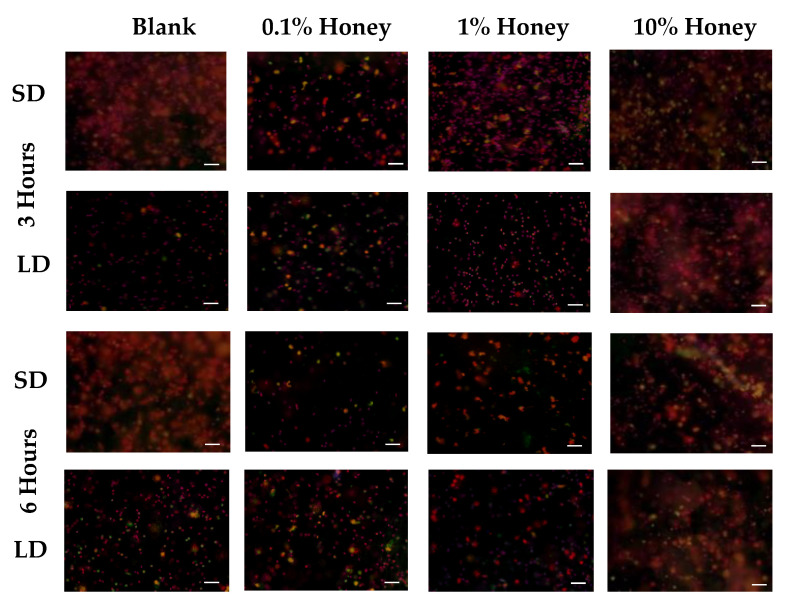
Representative fluorescence images of NETosis on LD and SD templates with 0%, 0.1%, 1%, and 10% honey at 3 and 6 h of culture, taken from experiment 1. Red indicates extracellular DNA (NETs), blue indicates intact nuclei, and green NE. Scale bar = 50 µm. Quantification is shown in Figure 5. Raw data is available in Appendix A, and single-channel images are available in Appendix A.

**Figure 5 polymers-12-01430-f005:**
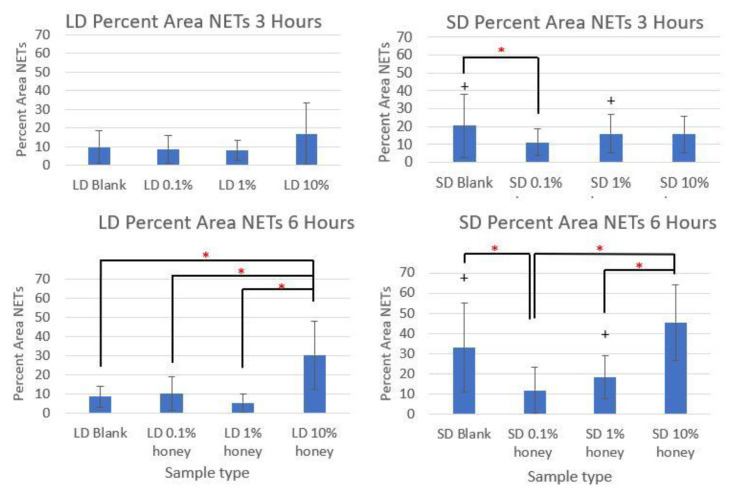
The incorporation of 0.1% honey significantly reduces NETosis on SD templates. percent area NETs on LD and SD templates at 3 and 6 h are shown. * indicates a significant difference between the indicated samples within that template type and timepoint, while + indicates a significant difference from the respective LD template at that timepoint and honey level. Sample size of 3 images per replicate, 4 replicates per timepoint per experiment, with 3 unique donor experiments. Raw data is available in Appendix A. Matlab code is available in Appendix A.

**Figure 6 polymers-12-01430-f006:**
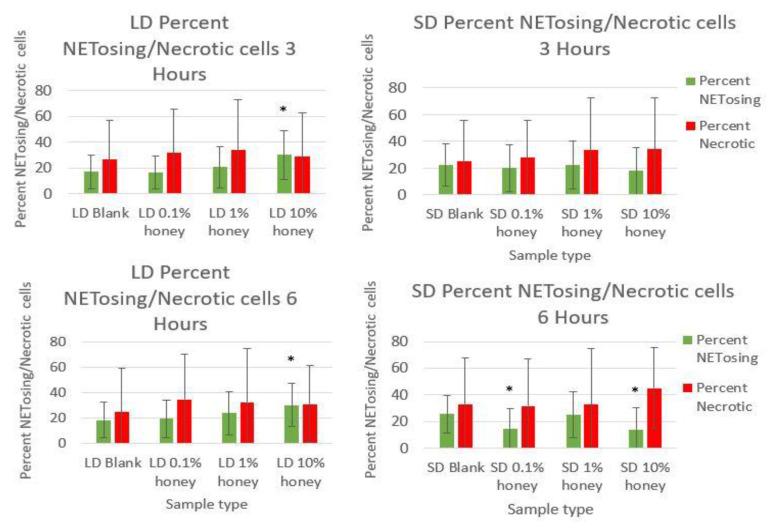
The reduction in NETosis is not caused by honey-driven necrosis. The percentage of cell nuclei identified as NETosing or necrotic are shown for LD and SD templates at 3 and 6 h. * indicates a significant difference from the respective LD or SD blank. Sample size of 3 images per replicate, 4 replicates per timepoint per experiment, with 3 unique donor experiments. Raw data is available in Appendix A.

**Figure 7 polymers-12-01430-f007:**
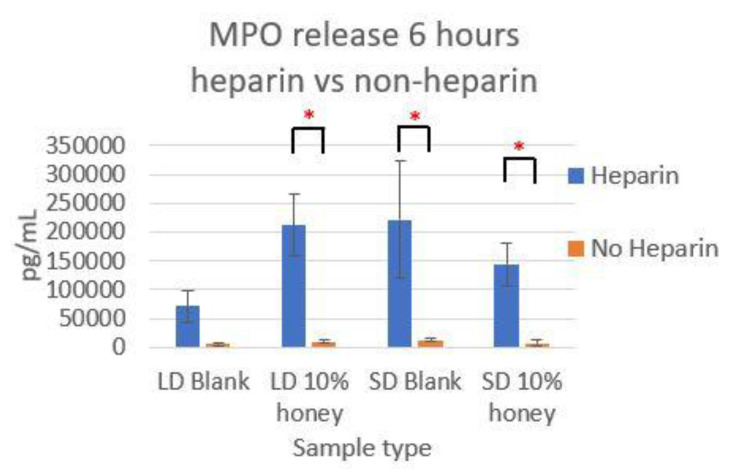
The release of non-NET-associated MPO is negligible. The concentration of MPO in the supernatant is shown for each sample cultured with or without the presence of heparin to free the MPO from the NETs. * indicates a significant difference between the indicated samples. Sample size of 3 replicates. Raw data is available in Appendix A.

**Figure 8 polymers-12-01430-f008:**
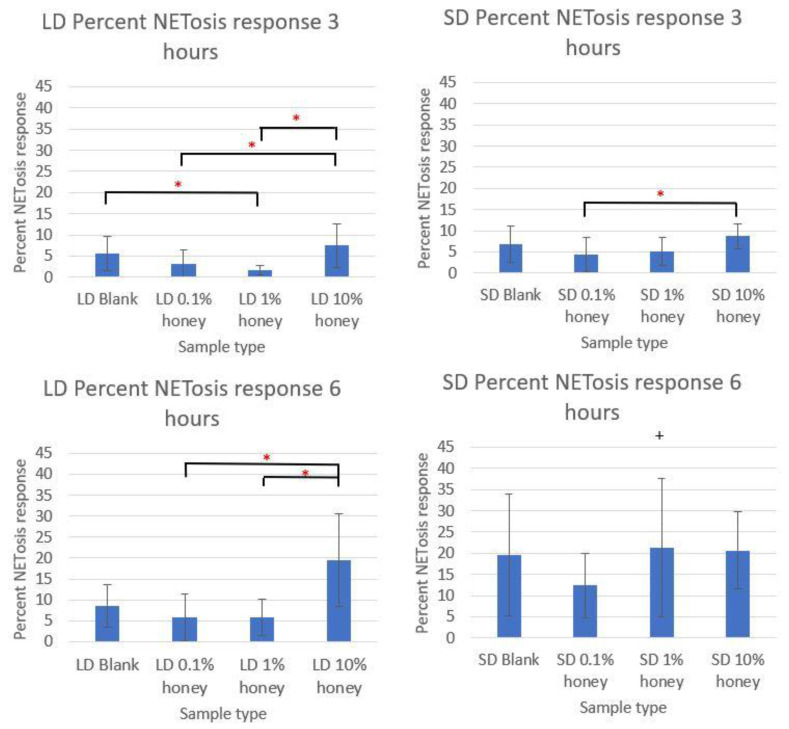
The 0.1% honey and 1% honey samples had the least NETosis response. The percent NETosis response, calculated from MPO concentration normalized to TCP 100 nM PMA positive controls, are shown on LD and SD templates at 3 and 6 h. * indicates a significant difference between the indicated samples within that template type and timepoint, while + indicates a significant difference from the respective LD template at that timepoint and honey level. Sample size of 4 replicates per timepoint per experiment, with 3 unique donor experiments. Raw data is available in Appendix A.

**Figure 9 polymers-12-01430-f009:**
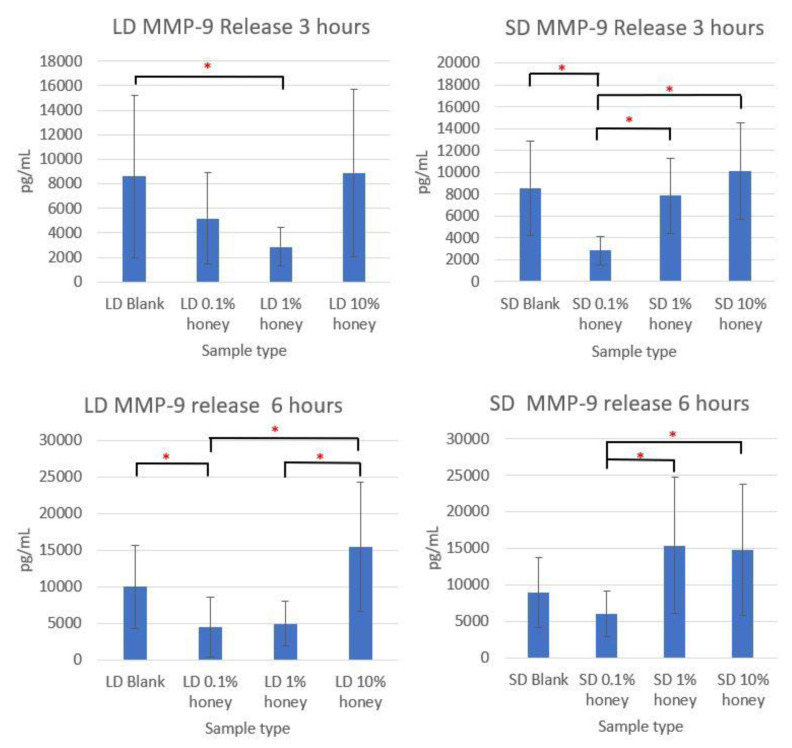
The 0.1% honey and 1% honey samples had the lowest MMP-9 release. The release of MMP-9 is shown on LD and SD templates at 3 and 6 h. * indicates a significant difference between the indicated samples within that template type and timepoint. Sample size of 4 replicates per timepoint per experiment, with 3 unique donor experiments. Raw data is available in Appendix A.

**Table 1 polymers-12-01430-t001:** Electrospinning parameters for each template type at low humidity conditions. Due to fluctuating humidity in the laboratory throughout this study, two sets of parameters had to be used to maintain consistent fiber diameters for each template type.

Electrospinning Parameters for SD and LD Templates under Low Humidity (<20%) Conditions					
Fiber Diameter	PDO Concentration (mg/mL)	Honey Concentration (% *v/v*)	Air Gap (cm)	Flow Rate (mL/h)	Voltage (+kV)
SD	67	0	17	0.25	18
		0.1	13		14
		1	13		14
		10	20		22
LD	135	0	28	4	25
		0.1			
		1			
		10			
		1			
		10			

**Table 2 polymers-12-01430-t002:** Electrospinning parameters for each template type at high humidity conditions. Due to fluctuating humidity in the laboratory throughout this study, two sets of parameters had to be used to maintain consistent fiber diameters for each template type.

Electrospinning Parameters for SD and LD Templates under High Humidity (> 20%) Conditions					
Fiber Diameter	PDO Concentration (mg/mL)	Honey Concentration (% *v/v*)	Air Gap (cm)	Flow Rate (mL/h)	Voltage (+kV)
SD	67	0	13	0.5	12
		0.1			
		1			
		10			
LD	140	0	18	4	25
		0.1			
		1			
		10			

**Table 3 polymers-12-01430-t003:** Fiber diameters and pore diameters of each template type (mean ± standard deviation). * indicates a significant difference between the pore diameters of the LD and SD templates at each honey condition. # indicates a significant difference from the non-honey blank with the corresponding fiber size (SD or LD). Three images of each template type were analyzed with a minimum of 200 fiber diameter measurements and 60 pore diameter measurements taken per image. Raw data is available in Appendix A.

Template Type	Fiber Diameter (μm) ± Std. Dev	Pore Diameter (μm) ± Std. Dev
SD Blank	0.49 ± 0.20	7.4 ± 4.6 *
SD 0.1% Honey	0.43 ± 0.11	4.3 ± 3.4 * #
SD 1% Honey	0.36 ± 0.13	3.0 ± 2.2 * #
SD 10% Honey	0.49 ± 0.23	2.6 ± 1.4 * #
LD Blank	1.75 ± 1.11	14.4 ± 8.9
LD 0.1% Honey	1.93 ± 1.38	17.2 ± 12.6
LD 1% Honey	2.17 ± 0.61	10.7 ± 5.7
LD 10% Honey	2.05 ± 1.07	9.3 ± 5.9

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
