# Peer review of "Manuka Honey Reduces NETosis on an Electrospun Template Within a Therapeutic Window"

_polymers, 2020, doi:10.3390/polym12061430_

Round 1

Reviewer 1 Report

In the present paper Manuka honey is incorporated into electrospun scaffolds with the aim of reducing neutrophil NETosis. The "Introduction", which describes the state of the art and states the hypothesis clearly, is well written, Similarly, the methodology is described in detail in the "Materials and Methods" section. In general, the data suggest some benefits of adding 0.1% and 1% honey to the scaffolds. However, the different assays show inconsistent results. For example, the effect of honey in reducing NETosis seemed to be more beneficial for SD (small diameter) samples than for LD (large diameter) samples according to extracellular DNA staining (Figure 4 and Figure 5). However, considering the MMP-9 release, the effect is more pronounced in LD samples. Moreover, for blank samples, MMP-9 levels are similar for SD and LD samples, which is not supported by the clear differences observed in DNA staining between these two samples in Figure 4 and 5. Most probably, there are some complex mechanism that are being overlooked in the discussion. Thus, it is recommended to describe the observed results with care to avoid overstating.

In general, some data and the statistical analysis does not match with the information described along the manuscript. Checking the data and statistics is needed. Some specific comments are provided below:

  • In the abstract, provide the full name for NET and MPO the first time they are mentioned.
  • Provide the full name for LPS, fMLP, NE, PDO, HFP, etc. (and in general to other abbreviations with are not generally known) the first time they are mentioned in the manuscript.
  • In Figure 1, it is recommended to remove small automated scale bars and replace with new clear bars, to ensure that they are legible.
  • If possible, and considering that x axis is common for all the graphs, try to combine the graphs in Figure 2 in two single graphs (one for LD and the other one for SD). In this way, the readers will have a more general view of the situation.
  • In figure 3, due to the huge difference in honey release between the 10% and 0.1 and 1% samples, the release of the samples with lower amount of honey is difficult to see. Thus, a zoom out of the release for these samples (0.1 and 1%) by changing the y axis is necessary.
  • In figure 4, the nuclei are hardly identified. A supporting figure showing the channels separately (and then merged) would be very useful.
  • In Figure 8, authors claim that “…and both the 0.1% and 1% samples had significantly less percent NETosis than the LD 10% samples at this timepoint” (3 hours). However, the way the statistics are presented in Figure 8 do not show that. Please, check the data for this particular case (LD-3 hours) in Figure 8 (the way it is represented, it seems that there is no difference between the 10% and the 1%).
  • In Figure 8, significant differences are observed for LD samples. However, in figures 4 and 5, the benefits of honey were observed in SD samples. This inconsistency between data should be clearly justified.
  • In Figure 9, authors claim that “the 0.1% honey LD sample had significantly less MMP-9 release than the LD blank samples at 3 hours”. However, this is not shown in the graph. Indeed, the sample that show significant differences with respect to the blank is that loaded with 1%. Please, check the data in this graph.
  • Again in Figure 9, authors claim that “Similarly, at 3 hours both 0.1% and 1% honey SD samples had less MMP-9 than the SD 10% honey samples”. However, according to the Figure 9 and the statistics performed, there are no differences between the 10% and 1% samples. Definitely, data and statistics have to be carefully checked in this figure.
  • Authors claim that “the lack of TNF-α, IL-8, IL-1β, IL-10 and IL-1ra release by neutrophils seeded on these templates indicate that they are not activated into pro-inflammatory or anti-inflammatory behaviour”. The presence of a scaffold loaded with honey should induce some change in the release of these cytokines. The sensitivity of the test or the over-dilution of the supernatant may have played a major role in these results.
  • In the discussion, the authors claim that “…the ability of 0.1% and 1% honey-containing templates to reduce MMP-9 release in neutrophils…”. However, these benefits were only observed for the LD samples. In fact, the MMP-9 release in the SD samples loaded with 1% honey was slightly higher than in the control (blank SD sample) after 6 hours.

Author Response

Please download the attachment

Reviewer 2 Report

The manuscript entitled "Manuka Honey Reduces NETosis on an Electrospun Template Within a Therapeutic Window" by Minden-Birkenmaier et al presented that Manuka honey was incorporated into electrospun templates with large (1.7-2.2 μm) and small (0.25-0.5 μm) diameter fibers at concentrations of 0.1, 1, and 10%. They found that these data indicate a therapeutic window for Manuka honey incorporation into tissue engineering templates for the reduction of NETosis. Just a few issues I would like to point out here for the authors to revise their manuscript.

(1) 2.1 Electrospinning: Authors should clearly provide the information about the abbreviated name of PDO materials? Please explain why the air gap used in all samples are quite different? Is there any reason or criteria that will affect the choice of this value of air gap?

(2) 2.4 response of primary human peripheral blood neutrophils: Besides the investigation of primary human peripheral blood neutrophils response, authors should conduct the tests of blood clotting times and hemolytic assays to confirm the biocompatibility of the SD and LD templates obtained at concentrations of 0.1, 1, and 10% of Manuka honey.

(3) Figure 3: Authors should conduct the SEM imaging for confirming the morphology differences (structural stability) of the SD and LD templates obtained at concentrations of 10% of Manuka honey over the 6-h and 21-day release period. Please explain how to obtain the value of percent honey in solution? Please explain why the percent honey in solution is as low as 1.5-2% for 10% H SD over the 21-day release period?

(4) The author should use other solvents to extract all the honey from the all samples for confirming the exact total amount of honey wrapped in fiber mats.

(5) Figure 6: Authors should conduct the negative control experiments to confirm the effects in HSBB medium when adding the same amount of 0.1, 1, and 10% of Manuka honey.

Round 2

Reviewer 1 Report

The authors of this paper addressed most of the comments. However, the figures still need to be greatly improved:

  • Figure 1: as mentioned in a previous comment, the scale bars are difficult to see. It is necessary to create your own scale bars with bigger size.
  • Figure 4: scale bar appear here and there within this figure. This has to be corrected.
  • Figure 8: there is an orange line at the bottom of the graph that does not provide any information.
  • Figure 9: as for Figure 8, there is an orange line at the bottom of the graph that does not provide any information.

Reviewer 2 Report

The authors addressed all of my concerns and gave a satisfactory answered. I am pleased to endorse publication at this point in time.

Author Response

Thank you very much!